# Seroprevalence of Severe Fever with Thrombocytopenia Syndrome Virus in Small-Animal Veterinarians and Nurses in the Japanese Prefecture with the Highest Case Load

**DOI:** 10.3390/v13020229

**Published:** 2021-02-02

**Authors:** Yumi Kirino, Keita Ishijima, Miho Miura, Taro Nomachi, Eugene Mazimpaka, Putu Eka Sudaryatma, Atsushi Yamanaka, Ken Maeda, Takayuki Sugimoto, Akatsuki Saito, Hirohisa Mekata, Tamaki Okabayashi

**Affiliations:** 1Center for Animal Disease Control, University of Miyazaki, Miyazaki 889-2192, Japan; kirinoyumi@gmail.com (Y.K.); sakatsuki@cc.miyazaki-u.ac.jp (A.S.); mekata@cc.miyazaki-u.ac.jp (H.M.); 2Department of Veterinary Sciences, Faculty of Agriculture, University of Miyazaki, Miyazaki 889-2192, Japan; 3Department of Veterinary Science, National Institute of Infectious Diseases, Tokyo 162-8640, Japan; keishi@nih.go.jp (K.I.); kmaeda@nih.go.jp (K.M.); 4Department of Microbiology, Miyazaki Prefectural Institute for Public Health and Environment, Miyazaki 889-2155, Japan; miura-miho@pref.miyazaki.lg.jp (M.M.); sugimoto-takayuki@pref.miyazaki.lg.jp (T.S.); 5Miyazaki Prefectural Miyakonojo Inspection Center, Miyazaki 885-0021, Japan; nomachi-taro@pref.miyazaki.lg.jp; 6Graduate School of Medicine and Veterinary Medicine, University of Miyazaki, Miyazaki 889-2192, Japan; emazimpaka1986@gmail.com (E.M.); putueka.sudaryatma@gmail.com (P.E.S.); 7Department of Internal Medicine, Miyazaki Prefectural Miyazaki Hospital, Miyazaki 880-8510, Japan; ayaman_555@yahoo.co.jp

**Keywords:** cats, dogs, SFTS, Bandavirus, public health, Japan

## Abstract

Severe fever with thrombocytopenia syndrome virus (SFTSV) is the causative agent of SFTS, an emerging tick-borne disease in East Asia, and is maintained in enzootic cycles involving ticks and a range of wild animal hosts. Direct transmission of SFTSV from cats and dogs to humans has been identified in Japan, suggesting that veterinarians and veterinary nurses involved in small-animal practice are at occupational risk of SFTSV infection. To characterize this risk, we performed a sero-epidemiological survey in small-animal-practice workers and healthy blood donors in Miyazaki prefecture, which is the prefecture with the highest per capita number of recorded cases of SFTS in Japan. Three small-animal-practice workers were identified as seropositive by ELISA, but one had a negative neutralization-test result and so was finally determined to be seronegative, giving a seropositive rate of 2.2% (2 of 90), which was significantly higher than that in healthy blood donors (0%, 0 of 1000; *p* < 0.05). The seroprevalence identified here in small-animal-practice workers was slightly higher than that previously reported in other high-risk workers engaged in agriculture and forestry in Japan. Thus, enhancement of small-animal-practice workers’ awareness of biosafety at animal hospitals is necessary for control of SFTSV.

## 1. Introduction

Severe fever with thrombocytopenia syndrome (SFTS) is a fatal, emerging, tick-borne zoonosis in East Asia. SFTS shows nonspecific clinical symptoms including high fever, strong fatigue, gastrointestinal symptoms, and leukocytopenia [1]. The SFTS mortality rate ranged from 10 to 27% in endemic countries such as China, Japan, and South Korea [2,3,4]. The causative agent of SFTS is *Dabie bandavirus*, which is generally known as SFTS virus (SFTSV), and is classified in the family *Phenuiviridae* and the genus *Bandavirus*. SFTSV is thought to be maintained among ticks and a range of host animals in the wild, and is sporadically transmitted to humans through tick bites [5]. At the time of discovery of SFTSV in Japan, people engaged in agriculture and forestry were considered to be at risk of infection, as with cases in China [1]. Recently, many cases of SFTS in domestic cats were reported in western Japan [6] and close contact with companion animals is also thought to be an important risk factor for SFTSV infection [3], suggesting that other workers might also be at occupational risk.

As of May 2020, in Japan, 517 confirmed cases of SFTS had been reported. Miyazaki prefecture, which is located in the south-east of Kyushu island (Figure 1), has the highest per capita number of recorded cases of SFTS in Japan, with 13.9% (72 of 517) of the total SFTS cases among only 0.8% of the country’s population.

We have previously identified direct SFTSV transmission from an infected cat to a veterinarian and a veterinary nurse at a small-animal hospital in Miyazaki prefecture [7]. Similar cases have also been reported in other SFTS-endemic prefectures [8], indicating that small-animal-practice workers, including veterinarians and veterinary nurses and assistants, might be at high risk of SFTSV infection. Our objective in this study was to determine the occupational risk of SFTSV infection among small-animal veterinarians and veterinary nurses, in order to contribute to the improvement of public health by enhancing awareness of those who are at risk.

## 2. Materials and Methods

### 2.1. Ethical Statement and Sample Collection

All protocols and procedures were approved by the research ethics committee of the Miyazaki Prefectural Institute for Public Health and Environment (Approval No. 2 on 23 May 2017).

In November 2018, 90 serum samples were collected from small-animal veterinarians (*n* = 43) and nurses (*n* = 47) working in Miyazaki prefecture. All participants were informed of the purpose of the study and gave written consent prior to blood sampling. A structured questionnaire was prepared and completed by the participants to collect information on job history; usage of personal protective equipment such as disposable gloves, mask, face shield, and goggles during medical examinations with animals; experience of contact with animals confirmed and/or suspected of having SFTS; and history of SFTS-like symptoms. The respondents in this study did not include the veterinarian and the veterinary nurse who were involved in the previously reported cat-to-human direct-transmission case [7]. The control group consisted of 1000 serum samples collected by the Japan Red Cross Society. All samples were provided according to the guideline of the effective use of donated blood for research and development (Acceptance No. 30J0025). The donors were healthy 18–64-year-old residents of Miyazaki prefecture who had been confirmed as antibody-negative for human immunodeficiency virus, hepatitis B and C viruses, human T-cell leukemia virus type 1, primate erythroparvovirus 1, and *Treponema pallidum*. 

### 2.2. Enzyme-Linked Immunosorbent Assay and Focus-Reduction Neutralization Test

The indirect enzyme-linked immunosorbent assay (ELISA) to detect human IgG antibodies against SFTSV antigens was performed as described previously [9]. Serum samples were diluted 100 times with PBS containing 5% Block Ace (Dainihon Seiyaku Co., Osaka, Japan) and 0.05% Tween-20. The value for detection by ELISA was calculated by the OD_405_ in wells coated with lysates of virus-infected cells minus the OD_405_ in wells coated with lysates of mock-infected cells, and the cut-off for a positive identification was set as 0.3 in this study. For any serum sample with an ELISA result greater than the cut-off value, further testing was performed with a focus-reduction neutralization test (FRNT) to quantify the FRNT_50_ value against SFTSV, as described previously [10]. The FRNT_50_ value is the reciprocal of the highest serum dilution that reduces the number of foci to 50% of the number in control wells. A FRNT_50_ value ≥10 was considered to indicate a seropositive sample.

### 2.3. Statistical Analysis

The Fisher’s exact test was used for comparison of the numbers of seropositive and seronegative samples between the small-animal-practice workers and blood donors. The analysis was performed using GraphPad Prism 6 software (GraphPad Software, San Diego, CA, USA). *p* < 0.05 was considered statistically significant in this study.

## 3. Results

Three serum samples (3.3%, 3 of 90), which were obtained from two veterinarians and a veterinary nurse working at small-animal hospitals, exceeded the cut-off value of the ELISA test (Table 1). 

The corrected OD_405_ values of these sera (sample IDs #4, #126, and #127) were 2.77, 3.17, and 1.56, respectively (Table 2). 

The FRNT_50_ values of the sera were 10, >160, and <10, respectively, so serum sample #127 was classified as seronegative, and the seropositive rate in the studied group of small-animal-practice workers in Miyazaki prefecture was 2.2% (2 of 90). By comparison, no seropositive cases were identified among 1000 serum samples obtained from blood donors in Miyazaki prefecture. The seropositive rate among the small-animal-practice workers was significantly higher than that among the blood donors (Fisher’s exact test, *p* = 0.0067).

According to the data gathered by questionnaire, a veterinarian who was seropositive (#126) and a veterinary nurse who was seronegative by FRNT, but seropositive by ELISA (#127), had both had previous contact with animals with SFTS-like presentation and had developed subjective SFTS-like symptoms. The other seropositive veterinarian (#4) had also had contact with animals with SFTS-like presentation but did not recall having noticeable symptoms indicating SFTS. Although the response rates for questions regarding the usage of personal protective equipment varied, 85.0% (57 of 67) and 86.3% (57 of 66) of small-animal-practice workers who responded to these questions indicated that they wore disposable gloves and masks, respectively, during clinical examinations. However, only 22.6% (12 of 53 respondents) indicated that they wore goggles or face shields. 

## 4. Discussion

To determine the occupational risk of SFTSV infection, we conducted a serological survey with samples from small-animal-practice workers and healthy blood donors in the prefecture with the highest per capita case load of SFTS in Japan. We found that 2.2% (2 of 90) of small-animal-practice workers were seropositive, indicating prior infection with SFTSV. By comparison, no seropositive samples were identified in the sera from 1000 healthy blood donors in the prefecture. The seropositive rate among small-animal-practice workers was significantly higher than among healthy blood donors (*p* = 0.0067). In this study, no seropositive case was confirmed among 1000 blood samples. Statistically, even if no seropositive case in more than 312 blood donors or two seropositive cases in 1000 blood donors had been confirmed, the significant difference of the seropositive rate among small-animal-workers was still recognized. In previous studies conducted in Japan, SFTSV seroprevalence was found to be 0.14% (1 of 694) among people >50 years old in Ehime prefecture, and 0.8% (1 of 125) among hunters in Kagoshima prefecture [11,12]. Although our sample size was limited, the seropositive rate that we identified among small-animal-practice workers was slightly higher than those found in other high-risk groups in Japan. 

In a study involving experimental infection of cats, high copy numbers of SFTSV were found in the serum, eye swabs, saliva, rectal swabs and urine [10]. As cat-to-human SFTSV transmission has been reported [7,8], small-animal practice should be considered as one of the occupations that is at high risk of SFTSV infection. Most of the veterinary workers who responded to the questionnaire in our study indicated that they wore disposable gloves and masks during clinical examinations (85.0–86.3%). However, only 22% wore face shields or goggles, suggesting that biosafety dissemination and awareness raising should focus on the wearing of face shields or goggles in addition to other basic personal protective equipment such as gloves, masks, and surgery gowns when handling cats and dogs suspected of having SFTS. 

The results of serological surveys are affected by the detection methods that are used. ELISA is a simple and high-throughput method, but nonspecific reactions may result in overestimation of the positive rate. Therefore, confirmatory virus-neutralization tests are considered desirable after positive results of screening by ELISA in serological surveys of SFTSV [11,12]. Here, we identified one case with a positive result by ELISA but a negative result by neutralization test. Although it is possible that a nonspecific reaction resulted in a false-positive detection by ELISA, another possibility is that the decay of SFTSV-neutralizing antibodies may have led to a false-negative result by FRNT. This veterinary nurse had experience of direct contact with a dog with SFTS-like presentation more than 10 years ago and had developed SFTS-like symptoms after the contact. Neutralizing antibodies against SFTSV have been shown to be maintained for >4 years post-infection [13]. Further research to follow the neutralizing-antibody profiles over time in relation to initial antibody titers will improve the precision of serological studies of SFTSV.

Our results have provided further evidence that small-animal-practice workers are at risk of occupational exposure to SFTSV. We have discussed this risk previously in relation to a case report of direct transmission of SFTSV from a cat to veterinary workers at a small-animal hospital [7]. The results of these studies demonstrate the urgent need for the establishment of veterinary standard precautions and improved biosafety education for veterinarians and veterinary nurses. 

## Figures and Tables

**Figure 1 viruses-13-00229-f001:**
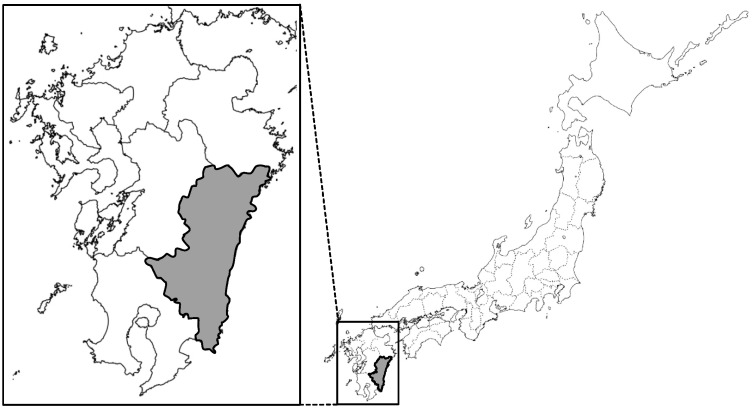
Map of Japan showing Miyazaki prefecture. The map indicates Miyazaki prefecture, where the sero-epidemiological survey was performed.

**Table 1 viruses-13-00229-t001:** Seroprevalence of severe fever with thrombocytopenia syndrome virus in Miyazaki prefecture, Japan.

Group	No. of ELISA-Positive Sera (Total No. of Tests)	No. of FRNT-Positive Sera (Total No. of Tests)	Seropositive Rate (No. Seropositive/Total No. Tested)
Small-animal-practice workers	3 (90)	2 (3)	2.2% (2/90)
Blood donors	0 (1000)	ND	0% (0/1000)

ELISA, enzyme-linked immunosorbent assay; FRNT, focus-reduction neutralization test; ND, not done.

**Table 2 viruses-13-00229-t002:** Characteristics of the serological reaction to severe fever with thrombocytopenia syndrome virus and the results from the questionnaire.

Participant ID	Occupation	ELISA OD_405_ Value (Test Minus Control) ^a^	FRNT_50_ Value ^b^	Experience of Contact with Animals with SFTS-Like Presentation	Experience of SFTS-Like Symptoms
#4	Veterinarian	2.27	10	Yes	No
#126	Veterinarian	3.17	>160	Yes	Yes
#127	Veterinary nurse	1.56	<10	Yes	Yes
ELISA negative veterinarians and veterinary nurses		-	-	Yes: 41.3% (26/63 ^c^)	Yes: 1.6% (1/63 ^c^)

^a^ Test wells were coated with lysates of SFTS-infected cells, and control wells were coated with lysates of mock-infected cells. ^b^ The FRNT_50_ value is the reciprocal of the highest serum dilution that reduces the number of foci to 50% of the number in control wells. ^c^ Number of valid responses other than the above three cases was 63. ELISA, enzyme-linked immunosorbent assay; FRNT, focus-reduction neutralization test; SFTS, severe fever with thrombocytopenia syndrome.

## Data Availability

The data presented in this study are available on request from the corresponding author. The data are not publicly available due to ethical restrictions.

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
