# Peer review of "Seroprevalence of Severe Fever with Thrombocytopenia Syndrome Virus in Small-Animal Veterinarians and Nurses in the Japanese Prefecture with the Highest Case Load"

_viruses, 2021, doi:10.3390/v13020229_

Round 1
Reviewer 1 Report
This manuscript described seroprevalence of severe fever with thrombocytopenia syndrome virus in small-animal veterinarians and nurses in the Japanese prefecture. The data suggested that occupational risk of SFTSV infection exist among small-animal-practice workers, and that personal protective equipments are required protective measures for related occupational workers when handling animals. This manuscript is well prepared, and can be accepted to be published in Viruses with minor language changes.
Author Response
Comments: Rev.1
This manuscript described seroprevalence of severe fever with thrombocytopenia syndrome virus in small-animal veterinarians and nurses in the Japanese prefecture. The data suggested that occupational risk of SFTSV infection exist among small-animal-practice workers, and that personal protective equipments are required protective measures for related occupational workers when handling animals. This manuscript is well prepared, and can be accepted to be published in Viruses with minor language changes.
>>Response to Reviewer1’s comment: We are thankful for the time and energy you expended. Following reviewer 1’s comments, we have corrected our English language as below;
Line36: rate is 10–27%>>ranged from 10-27%
Line43: However1>>Recently
Line65: in >>among
Reviewer 2 Report
The authors present a well written sero-survey of SFTSV in Japan. The intent of the manuscript is to determine if veterinarians and their staff have a higher seroprevalence of SFTSV antibody than the normal population. The authors compare 99 veterinarians and staff and found a 3% positivity rate. In a collection of 1,000 blood donor samples no positivity was found. A Fisher's test indicated statistical significance. The data support low seroprevalence of SFTSV in the at-risk population, and likely exceedingly rare positivity in the general population. The only concern is the large number of normal blood donors and statistical test used. If a smaller number of normal blood donors was used, would the significance have altered? Is the positivity rate significant due to the large number of normal samples? A few sentences explaining the appropriateness of the Fisher's test would help the manuscript.
Author Response
Rev. 2
The authors present a well written sero-survey of SFTSV in Japan. The intent of the manuscript is to determine if veterinarians and their staff have a higher seroprevalence of SFTSV antibody than the normal population. The authors compare 99 veterinarians and staff and found a 3% positivity rate. In a collection of 1,000 blood donor samples no positivity was found. A Fisher's test indicated statistical significance. The data support low seroprevalence of SFTSV in the at-risk population, and likely exceedingly rare positivity in the general population. The only concern is the large number of normal blood donors and statistical test used. If a smaller number of normal blood donors was used, would the significance have altered? Is the positivity rate significant due to the large number of normal samples? A few sentences explaining the appropriateness of the Fisher's test would help the manuscript.
>>Response to Reviewer2’s comment:
Following the reviewer’s comment, we added two sentences as follows.
Line 144: In this study, no seropositive case was confirmed among 1,000 blood samples. In statistically, even if no seropositive case in more than 312 blood donors or two seropositive cases in 1,000 blood donors had been confirmed, the significant difference of the seropositive rate among small-animal-workers was still recognized.
Reviewer 3 Report
Severe fever with thrombocytopenia syndrome virus (SFTSV) causes a severe, often fatal, disease SFTS in humans. Potential transmission from small companion animals, cat and dogs, are recently recognized as important public concern. Kirino et al. investigated the seroprevalence of SFTSV in veterinarians and veterinary nurses who work in SFTSV endemic area, Miyazaki prefecture. The authors demonstrated higher seropositive rate in veterinarians and veterinary nurses compared to healthy control. These data are important information for public health.
The manuscript seems to be worthy to be published, however, several minor points should be addressed before that.
Major points
In this manuscript, the authors collected information about job history; usage of personal protective equipment, experience of contact with animals confirmed and/or suspected of having SFTS, and history of SFTS like symptoms. If this information can be open, depend on the agreement, it is important for discussion with seropositive rate in veterinarians and veterinary nurses. Other participants who have shown negative serologic reactions may also have a history of SFTS-like symptoms. These data (% of contact experience, % of SFTS like symptoms etc.) should be shown as Table 1, more important data than Figure 1.
Minor points
Introduction: Cases of SFTSV infection in cats and dogs also need introduce in introduction section to indicate the high potential for contact with veterinarians.
L43, "Additionally" or "Recently" may be more suitable than "However" for this sentence.
L85, Sample serum preparation and serum dilution should be shown in the methods.
L104-105, This sentence is a text of results or in table?
L114, Between table and text need space.
L158-160, The authors discussed about limitation of methods, both ELISA and FRNT. In this study, #127 showed the results as ELISA-positive and FRNT-negative even though who had previous contact with animals with SFTS-like presentation and had developed SFTS-like symptoms. This is not a case report, however, the authors need to discuss about this data as well as other negative data if they had contact history with animals with SFTS-like presentation or had potential SFTSV infection history.
Author Response
Major points
In this manuscript, the authors collected information about job history; usage of personal protective equipment, experience of contact with animals confirmed and/or suspected of having SFTS, and history of SFTS like symptoms. If this information can be open, depend on the agreement, it is important for discussion with seropositive rate in veterinarians and veterinary nurses. Other participants who have shown negative serologic reactions may also have a history of SFTS-like symptoms. These data (% of contact experience, % of SFTS like symptoms etc.) should be shown as Table 1, more important data than Figure 1.
>>Response to Reviewer3’s comment: Following reviewer’s comments, we have added the data (% of contact experience with SFTS-like animals, % of SFTS like symptoms) of other participants who have shown negative serologic reactions in Table 2.
Minor points
Introduction: Cases of SFTSV infection in cats and dogs also need introduce in introduction section to indicate the high potential for contact with veterinarians.
L43, "Additionally" or "Recently" may be more suitable than "However" for this sentence.
>> Following reviewer’s pointed, we have corrected it.
L85, Sample serum preparation and serum dilution should be shown in the methods.
>> Following reviewer’s pointed, we have corrected it at Line 87
L104-105, This sentence is a text of results or in table?
L114, Between table and text need space.
>> We put spaces between Table and text at Line 105, 108, and 119.
L158-160, The authors discussed about limitation of methods, both ELISA and FRNT. In this study, #127 showed the results as ELISA-positive and FRNT-negative even though who had previous contact with animals with SFTS-like presentation and had developed SFTS-like symptoms. This is not a case report, however, the authors need to discuss about this data as well as other negative data if they had contact history with animals with SFTS-like presentation or had potential SFTSV infection history.
>> As reviewer suggested, we had added the information of case #127 as follow; “This veterinary nurse had experience of direct contact with a dog with SFTS-like presentation more than 10 years ago and had developed SFTS-like symptoms after the contact.” at Line 172.